# Metal-rich stars are less suitable for the evolution of life on their planets

**Anna V. Shapiro** [1] ✉, **Christoph Brühl**[2], **Klaus Klingmüller** [2], **Benedikt Steil**[2], **Alexander I. Shapiro** [1], **Veronika Witzke**[1], **Nadiia Kostogryz**[1], **Laurent Gizon** [1,3,4], **Sami K. Solanki** [1,5] **& Jos Lelieveld** [2,6]

Atmospheric ozone and oxygen protect the terrestrial biosphere against harmful ultraviolet (UV) radiation. Here, we model atmospheres of Earth-like planets hosted by stars with near-solar effective temperatures (5300 to 6300 K) and a broad range of metallicities covering known exoplanet host stars. We show that paradoxically, although metal-rich stars emit substantially less ultraviolet radiation than metal-poor stars, the surface of their planets is exposed to more intense ultraviolet radiation. For the stellar types considered, metallicity has a larger impact than stellar temperature. During the evolution of the universe, newly formed stars have progressively become more metal-rich, exposing organisms to increasingly intense ultraviolet radiation. Our findings imply that planets hosted by stars with low metallicity are the best targets to search for complex life on land.

Complex, multicellular life on land requires oxygen ($O_2$) from which ozone ($O_3$) forms[1], leading to a tolerable ultraviolet radiation (UV) level at the surface for its development and evolution[2–4]. Stellar emission and planetary UV protection depend on the effective temperature of the host star[5–7]. While for a young planet UV exposure can be essential for abiogenesis[8–11], high levels of UV trigger genomic damage and are a threat to all life forms[12–14]. In the Sun-Earth system the UV-C (202 to 230 nm, wavelengths potentially reaching the surface in oxygenated atmospheres) and UV-B (280 to 315 nm) fluxes at 1 Astronomical Unit (au) from the Sun are about 0.76 W/m² and 20 W/m², respectively[15]. This is well above the maximum tolerable level for terrestrial life. Land-based life has nevertheless evolved on Earth through oxygen enrichment of the atmosphere that blocks most of the UV radiation. While UV-C is largely absorbed by $O_2$ molecules in the upper atmosphere, UV-B is absorbed by the ozone layer in the middle atmosphere.

The $O_3$ concentration is regulated by a photochemical cycle of $O_2$ and $O_3$ dissociation by solar UV radiation in the Herzberg continuum (200 to 242 nm) and Hartley band (200 to 320 nm), respectively[1]. Only the longer wavelengths (260 to 320 nm) of the Hartley band are relevant for the $O_3$ column density burden because

radiation at shorter wavelengths is mostly absorbed by $O_2$. Ozone absorption in the band center (260 nm) is so strong that radiation can hardly penetrate to the middle atmosphere where $O_3$ concentrations are highest (Supplementary Fig. 1 and 3). Hence, the $O_3$ concentration depends on the balance between stellar irradiance in the 200 to 242 nm and 260 to 320 nm spectral bands, which govern the production and destruction of $O_3$, respectively (hereafter we refer to the net photochemical effect). Consequently, the UV-protection provided by the planetary atmosphere depends on the spectral distribution of the stellar radiation[5,7,16].

The stellar radiative spectrum, in turn, depends on the effective temperature, $T_{eff}$, and metallicity, [Fe/H], that represents the abundance of elements heavier than hydrogen and helium in a star (see Eq. 4 in methods section Stellar spectra). The dependence of the radiative conditions at the planetary surface on the stellar effective temperature has been studied previously. For example, it was shown that with increasing effective temperature the planetary surface UV above 290 nm also increases, but the radiative transfer at shorter wavelengths is non-monotonous due to spectrally dependent photochemical effects[5,7].

[1]Max Planck Institute for Solar System Research, Göttingen, Germany. [2]Max Planck Institute for Chemistry, Mainz, Germany. [3]Institute for Astrophysics, Georg-August-Universität Göttingen, Göttingen, Germany. [4]Center for Space Science, NYUAD Institute, New York University Abu Dhabi, Abu Dhabi, UAE. [5]School of Space Research, Kyung Hee University, Yongin, Republic of Korea. [6]The Cyprus Institute, Climate and Atmosphere Research Center, Nicosia, Cyprus. ✉e-mail: shapiro@mps.mpg.de

Here, we investigate the dependence of planetary surface UV on the atmospheric $O_2$ concentration and stellar metallicity for stars of three spectral types: G2V ($T_{eff}$ =5800 K, representing solar case), G5V ($T_{eff}$ =5300 K), and F7V ($T_{eff}$ =6300 K). We note that the G5V and F7V classes encompass roughly 50% of the presently known planetary hosts. We account for a range of metallicity values between −1 and 0.9 dex which covers most planetary hosts. We first consider the development of the Sun-Earth system in the past 0.5 billion years as a model for planets and their host stars, during which the atmosphere was oxygenated and complex life on land evolved. We then study the dependence of surface UV irradiation on the atmospheric $O_2$ content and stellar metallicity. We show that the development of complex life on planets in the habitable zone can be sustained from a few percent of $O_2$ upward, being robust for a large range of stellar characteristics and against major extraterrestrial cataclysms.

## Results

Figure 1 presents stellar spectra calculated for different metallicity values using the recent MPS-ATLAS code[17]. The UV flux drops substantially with increasing metallicity, creating seemingly more favourable conditions for life[18]. However, Fig. 1b shows that metallicity affects radiation in the $O_3$-producing Herzberg continuum much more strongly than in the $O_3$-destroying Hartley band. Thus, the net photochemical effect leads to a decrease of $O_3$ with metallicity, making the assessment of the UV conditions and potential habitability at the planetary surface less straightforward, which was hitherto not accounted for.

We consider hypothetical Earth-twin planets (with an $N_2/O_2$ atmosphere, water and terrestrial mass) in the habitable zones of stars

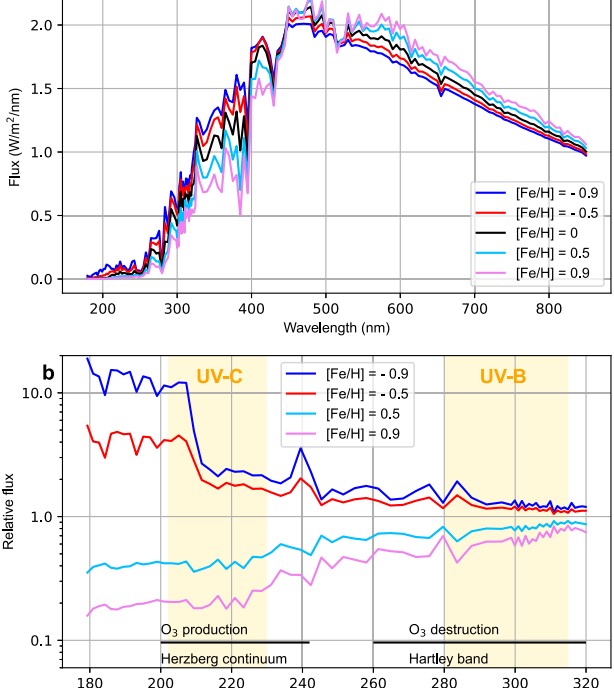

**Fig. 1 | Stellar radiative spectra. a** Stellar spectra calculated for different metallicities [Fe/H] and a solar effective temperature $T_{eff}$ of 5800 K. The calculations were performed for a spectral range of 170 to 850 nm and the total flux normalized to the solar constant. **b** The same as **a** but the flux is shown relative to the solar flux ([Fe/H] = 0). Radiation in the 200 to 242 and 260 to 320 nm intervals participates in $O_3$ production and destruction, respectively. These spectral ranges are defined by black solid lines. The yellow shaded areas show the spectral ranges of the UV-C reaching the lower atmosphere (202 to 230 nm, left) and UV-B (280 to 315 nm, right). Source data are provided as a Source Data file.

with $T_{eff}$ and [Fe/H] in ranges introduced above. The $O_3$ concentration in planetary atmospheres and the associated surface UV levels were computed with a coupled photochemical radiative-convective model[19]. The model has been designed and updated for accurate calculations of atmospheric $O_3$ chemistry. It avoids assumptions of fixed boundary conditions, e.g. of methane ($CH_4$) concentrations, and interactively computes the oxidation capacity of the atmosphere for different UV conditions, which is relevant for the removal of hazardous and greenhouse gases. The radiative scheme has a high spectral resolution, which appears to be critical for studying the impact of stellar radiation on planetary atmospheres[1]. The description of the planetary atmospheric model and our calculations of the input stellar spectra are presented in the methods sections Atmospheric model and Stellar spectra. We consider three key factors for a life-supporting UV environment on surfaces of planets: (1) UV-C level (relevant in low oxygen atmospheres); (2) the oxidation capacity of the lower atmosphere (regulated by UV radiation); (3) UV-B level (e.g., regarding DNA damage[2,3]). Further, the atmospheric protection mechanism should be able to tolerate large disturbances, e.g. by volcanic eruptions and supernova explosions (which can destroy $O_3$[20,21]) as well as changes of the activity of the host star[22] which affects its radiation spectrum[23] (see methods sections Ozone and UV-B response to perturbations and Stellar spectra).

### The Sun-Earth evolution

A quantitative assessment of the biological impact of these factors is challenging because additional protection (e.g. water bodies, shadowing by rocks, pigment formation) and biological repair mechanisms are not known. Therefore we considered the Earth-Sun system as a paradigm to guide the interpretation of our results for other systems. We first investigated how life on land of our planet steered through the conditions mentioned above and then how these conditions are affected by the effective temperature and metallicity of the host star.

Figure 2 illustrates the development of the Earth's atmosphere and surface UV fluxes over the last 600 Myr (million years before present). The geological isotope records indicate that the level of atmospheric $O_2$ (and $CO_2$, see Supplementary Table 1) went through substantial fluctuations[24–27] (Fig. 2a). The largest change in $O_2$, known as the Paleozoic oxygenation event, happened around 470 Myr[28] when Earth's atmosphere went from almost anoxic to oxygenated conditions. While this event is absent in the reconstruction of Berner et al.[24] (B6, blue curve in Fig. 2a) data (probably because of simplified assumptions about sulfur geochemistry[29]), it is evident in the more recent reconstruction of Lenton et al., 2016[27] (L16, light blue curve in Fig. 2a).

Before the Paleozoic oxygenation $O_2$ was mainly produced by aquatic photosynthesis in UV tolerant cyanobacteria and algae[26,30,31] which could only provide a limited amount of $O_2$: while L16 indicates 4% of $O_2$ shortly before the event, other studies point to lower values of 2%[32] or even 0.2%[26,33]. It is understood that the Paleozoic oxygenation event was caused by the advent of the earliest land plants[27,29]. This transition likely represents an important bottleneck: effective $O_2$ release to the atmosphere is not possible without land plants, which in turn are susceptible to UV radiation[33,34] and can only appear when the $O_2$ concentration is sufficiently high to create a protective ozone layer. We have modelled the history of the Earth's atmosphere to calculate the $O_2$ and $O_3$ concentrations needed to overcome the oxygenation bottleneck on Earth and simultaneously establish quantitative criteria on the non-hostile level of UV irradiance.

Here we consider the evolution of UV-C and UV-B fluxes at the surface. While UV-C is particularly harmful to living cells due to the highly energetic photons, Fig. 2b shows that a flux of $10^{-3}$ $Wm^{-2}$, corresponding to a tolerable annual dose in the order of $10^4$ $J\,m^{-2}$ [35], is achieved already at 0.3% of $O_2$. This is below the $O_2$ level after the Paleozoic oxygenation event. It is also below or comparable to

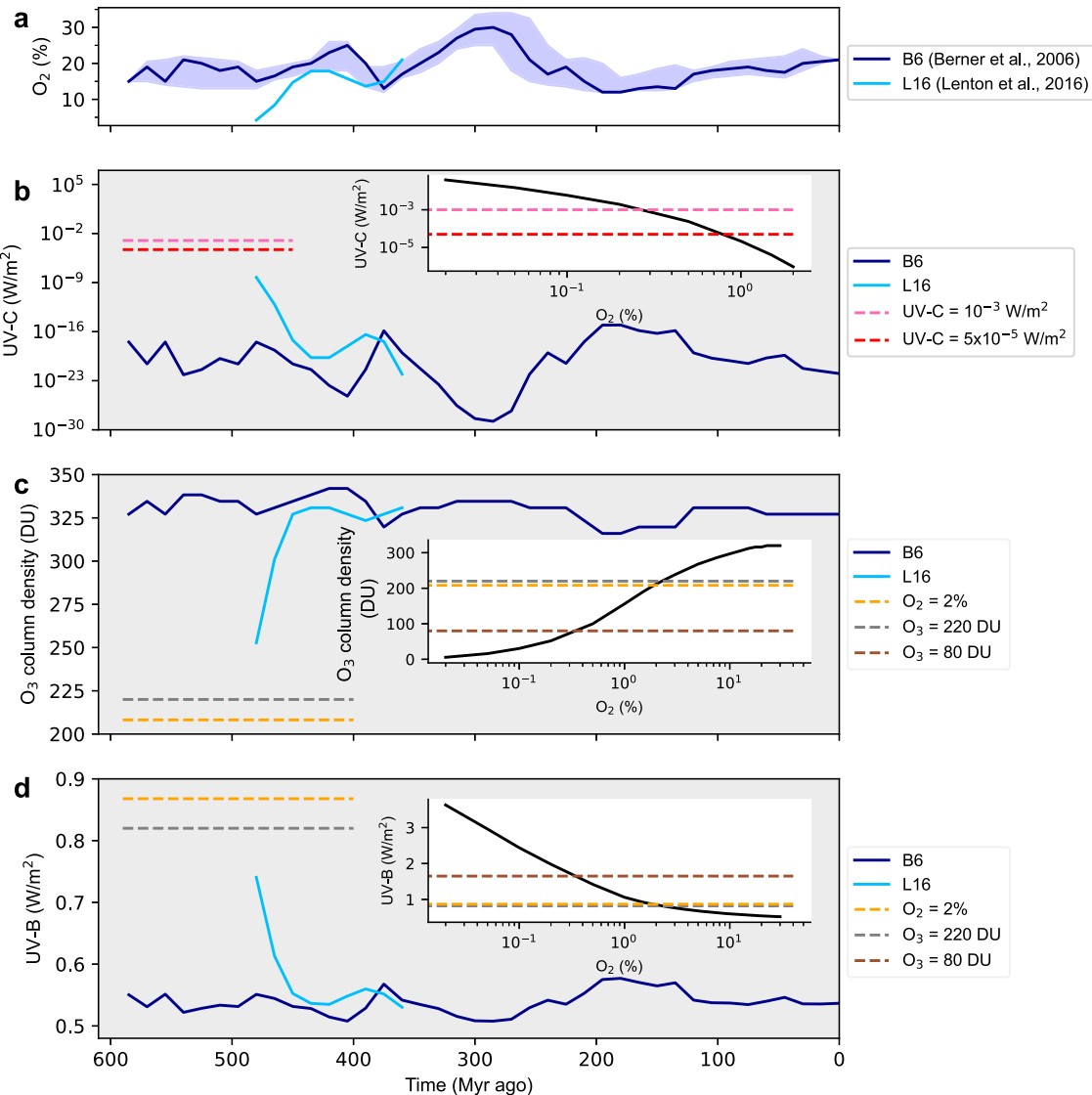

**Fig. 2 | Evolution of atmospheric O₂, O₃ column density and surface UV. a** O₂ volume fraction of the Earth's atmosphere during the past 600 million years according to Berner et al., 2006[24] (B6, blue) and Lenton et al.[27] (L16, light blue). The blue shaded area indicates the uncertainty range of the B6 data. **b–d** Surface UV-C (202 to 230 nm) **b**, O₃ column density **c** and surface UV-B (280 to 315 nm) **d** (1 DU = 2.687 × 10²⁰ molecules/m²). The blue and green curves are calculated with B6

and L16 data, respectively. The inserted plots show the dependencies of UV-C **b**, O₃ column density **c** and UV-B **d** on the O₂ content. **c**, **d** The level of O₃ corresponding to 2% of O₂ is shown by the yellow dashed line. The O₃ level of 220 and 80 DU are indicated by the grey and brown dashed lines, respectively. The UV-C levels of 10⁻³ and 5×10⁻⁵ W/m² are shown by the pink and red dashed lines in **b**. Source data are provided as a Source Data file.

available estimates for the O₂ levels preceding the event[27,32,33]. All in all, we do not expect the UV-C irradiance to pose a critical threat to the advent of land plants directly. Long CH₄ lifetimes resulting from our calculations, being a metric of the atmospheric oxidation capacity, indicate that the low UV-C values allow a mildly oxidative environment supporting the removal of hazardous gases in the lower atmosphere while not exposing organisms to harmful oxidant levels. It means that the atmosphere is not chemically aggressive, i.e. hostile to the organic molecules of living cells at UV-C < 5 × 10⁻⁵ W/m² or O₂ > 0.8% (Figs. 2b and 3e and Supplementary Fig. 5k). Under these more moderately oxidative conditions, the CH₄ lifetime exceeds about a year and thus does not drop below about a tenth of that in the atmosphere of today.

Since UV-B is mainly absorbed by O₃, the surface UV-B level must be calculated together with the atmospheric chemistry and O₃ concentration. We show the results of such calculations in Fig. 2c, d. Previous work[5] has shown that O₃ column density is resilient even to strong changes of O₂ (Fig. 2c). Our study reveals that the Paleozoic oxygenation event, which corresponds to a change in O₂ by a factor of

4.5 (L16 data), resulted in a mere 30% change of O₃ column density (from 252 to 330 Dobson Units, DU) leading to a moderate UV-B response (from 0.75 to 0.55 W/m², Fig. 2d). The low sensitivity of O₃ column density and, consequently of UV-B, to changes in O₂ is explained by the vertical adjustment of the ozone layer. Smaller O₂ amounts cause the ozone layer to form in lower and denser atmospheric layers where more O₂ is available for O₃ production (Supplementary Fig. 3a, b), leaving the total O₃ column density only weakly affected[36].

The total O₃ column density of 252 DU (Fig. 2c) calculated for the conditions shortly before the Paleozoic oxygenation event (L16 data) is well above the 220 DU (grey dashed line in Fig. 2c) that define the recent ozone hole over Antarctica[21] and can be considered tolerable by the vegetation. Since the O₂ concentration before the Paleozoic oxygenation event is rather uncertain we consider even lower oxygen levels. Lowering O₂ to 2% results in O₃ decrease to 205 DU (orange dashed line in Fig. 2c) and a UV-B increase to 0.87 W/m² (Fig. 2d). Though this amount of O₃ qualifies as an ozone hole, it is routinely

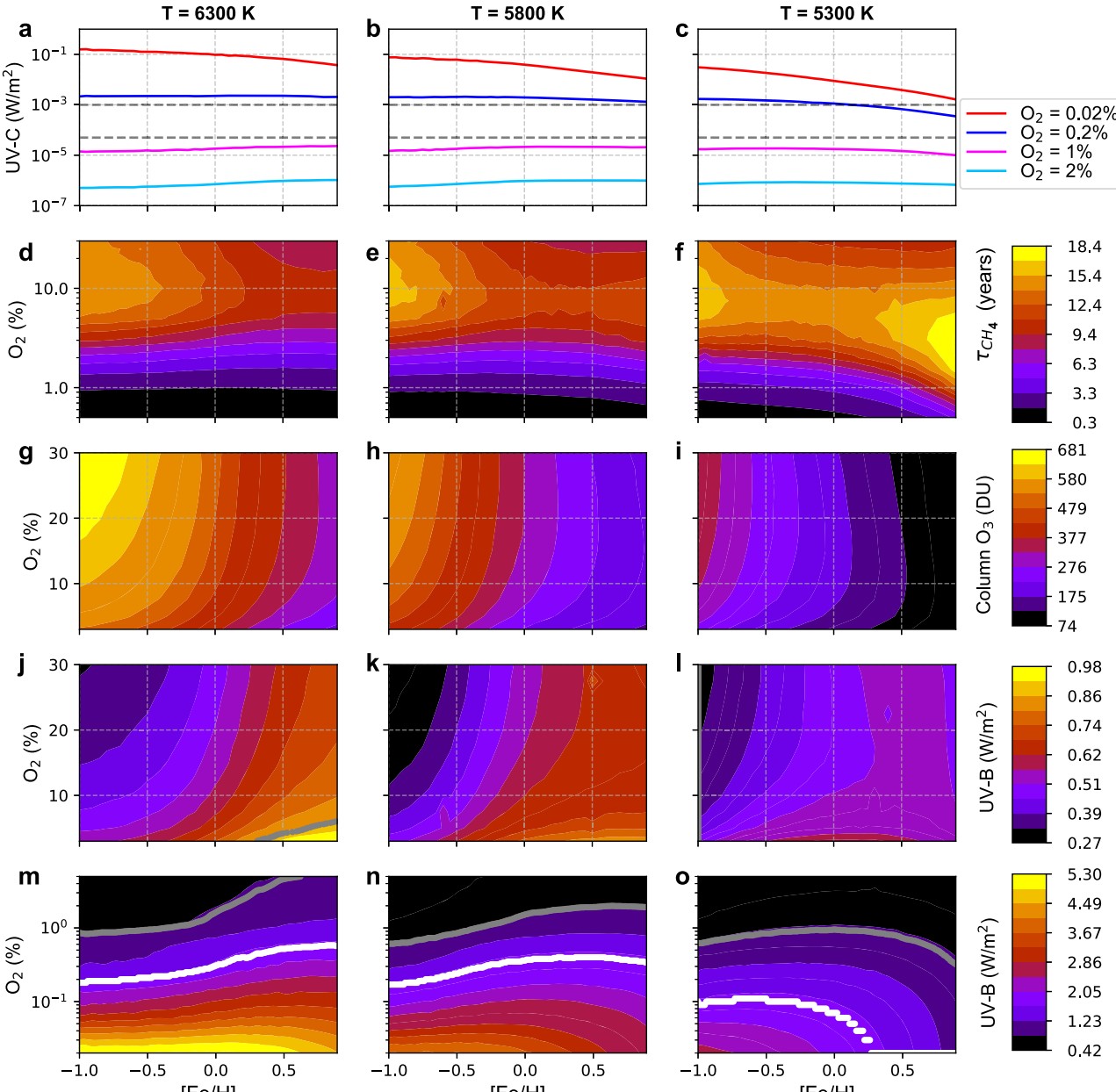

**Fig. 3 | Metallicity impacts on the atmosphere. a–c** The change of surface UV-C (202 to 230 nm, W/m²) with [Fe/H] for O₂ levels of 2% (light blue), 1% (magenta), 0.2% (blue) and 0.02% (red). The dashed lines indicate the UV-C levels of $5 \times 10^{-5}$ and $10^{-3}$ W/m². **d–f** Impact of O₂ content and [Fe/H] on CH₄ lifetime. **g–l** Dependencies of O₃ column density (**g–i** DU) and surface UV-B (**j–l** W/m²) on O₂ content and stellar metallicity [Fe/H]. **m–o** The same as j-l but for an O₂ content less than 5%. **a–o** The calculations were performed for stellar $T_{\mathrm{eff}}$ of 6300 K (left), 5800 K (center) and 5300 K (right). The grey and white curves (**j–o**) represent surface UV-B of 0.82 W/m² (220 DU, ozone hole definition) and 1.65 W/m² (80 DU, extreme in ozone hole), respectively. Source data are provided as a Source Data file.

measured over the Antarctic coastline[37] and sometimes over southern Argentina[38] or northern Europe where land plants are nevertheless abundant. Thus, also this level of UV-B does not pose a lethal threat to the terrestrial biosphere. The O₃ column density and UV-B modelled for O₂ below 2% are presented in the inserts of Fig. 2c, d. The O₃ column density of 80 DU (brown dashed line in Fig. 2c) that corresponds to O₂ of 0.3% is close to the lowest value measured over Antarctica in spring (September 30, 1994, with 73 DU) which was tolerable only due to the large solar zenith angle near the pole and the limited time period. Here, we consider the corresponding UV-B level of 1.65 W/m² (Fig. 2d) as the highest value that is known to be survivable by land plants on present Earth.

Our calculations thus show that the dependence of the surface UV-B fluxes on the O₂ amount is limited. We consider this to be a key

factor, together with the atmospheric oxidation capacity, for allowing land plants on Earth, which might similarly apply to habitable planets orbiting other stars. Furthermore, during the past 470 million years the dependence of the Earth's surface UV-B fluxes on the O₂ amount has been limited because the O₂ mixing ratio was well above 1%. Together with the atmospheric oxidation capacity, we consider this crucial for land plants on Earth, which might similarly apply to habitable exoplanets.

**Impacts of stellar properties**

The UV emission by a star strongly depends on fundamental stellar parameters. It increases with the stellar effective temperature $T_{\mathrm{eff}}$[5–7] (Supplementary Fig. 2) and decreases with the metallicity [Fe/H] (Fig. 1). We investigated how these parameters affect the

correspondence between $O_2$ in the planetary atmosphere and the surface UV thresholds indicated above. Note that $O_2$ in exoplanetary atmospheres may also be provided by abiotic sources[39,40]. Figure 3a–c shows that for 1% of $O_2$ the surface UV-C flux is lower than the level of $5 \times 10^{-5}$ W/m² for all stellar $T_{eff}$ and [Fe/H] values. We find that for 0.2% or more $O_2$ the higher UV-C fluxes of metal-poor stars are compensated by higher $O_3$ concentrations resulting in almost metallicity-independent UV-C at the planetary surface (Fig. 3a–c and Supplementary Fig. 4). This is remarkable given the strong anti-correlation of stellar UV-C emission fluxes and metallicity (Fig. 1b). Interestingly, the UV-C level also marginally depends on stellar $T_{eff}$. Thus, the $O_2$ level that provides UV-C protection is very similar for a large range of stellar parameters.

In the case of very low $O_2$ (< 0.7%), UV-C photolysis of $H_2O$ controls the formation of oxidants like excited O atoms and hydroxyl (OH) radicals (the latter known as the detergent of the atmosphere), which act as sinks for $CH_4$ and other molecules released from geochemical processes. Consequently the lifetime of $CH_4$ ($\tau_{CH_4}$), which signifies the oxidation capacity of the atmosphere (Fig. 3d–f), is very short (below 1 year), which likely makes the planetary environment harmful for life on land. For $O_2$ in excess of about 0.7% the OH concentration (Supplementary Fig. 5j–l) and $\tau_{CH_4}$ are controlled by UV-B photolysis of $O_3$, modulated by ambient concentrations of $H_2O$, $NO_x$ (mostly NO, $NO_2$) and CO. While surface $O_3$ for $O_2$ levels between 0.7 and 2% strongly depends on $T_{eff}$ and metallicity (Supplementary Fig. 5d–f), UV-B in the lower planetary atmosphere responds much less sensitively (Fig. 3m–o), and so does $\tau_{CH_4}$ (Fig. 3d–f). Above 2-3% $O_2$, the atmospheric oxidation capacity is buffered, hence stable and benign to life, maintaining a $\tau_{CH_4}$ within 8-18 years throughout the range of [Fe/H] and $T_{eff}$. For details about the temperature and atmospheric composition, see methods section Atmospheric model and Supplementary Fig. 5.

The effects of the stellar temperature $T_{eff}$ and metallicity [Fe/H] on UV are stronger in the Herzberg continuum, driving the chemical $O_3$ production, than in the Hartley band which drives $O_3$ destruction (see Fig. 1 for the [Fe/H] effect and Supplementary Fig. 2 for the $T_{eff}$ effect). As a result, the net photochemical effect leads to the increase of the $O_3$ column density with $T_{eff}$ and a decrease with [Fe/H] (Fig. 3g–i). Thus, changes in UV irradiance at the planetary surface can respond in the opposite direction to those of the stellar UV radiance. The net photochemical effect is not sufficiently strong to reverse the UV-B surface flux dependence on $T_{eff}$ for all metallicities and oxygen levels (Fig. 3j–l, Supplementary Fig. 2), but it generally reverses the dependence on [Fe/H] leading to the paradoxical anti-correlation of the surface and space UV-B fluxes (Figs. 1 and 3j–l). The increase of the surface UV-B with [Fe/H] is especially strong for $T_{eff}$ values of 5800 and 6300 K but it is less pronounced for 5300 K stars (where surface UV-B starts to decrease with [Fe/H] for low $O_2$ levels, see Fig. 3o).

The white lines in Fig. 3m, n indicate that for the $T_{eff}$ values of 5800 and 6300 K approximately two times more $O_2$ is needed to attain the extreme ozone hole conditions (i.e. UV-B flux of 1.65 W/m²) for [Fe/H] = 0.9 than for [Fe/H] = −1. The oxygen concentration required to arrive at lower UV-B levels such as ozone hole conditions (0.82 W/m², black lines in Fig. 3k–o) depends even more strongly on the metallicity for 5800 K and 6300 K stars. For example, for the effective temperature of the Sun the required $O_2$ concentration increases from 0.6% at [Fe/H] = −1 to about 2% for [Fe/H] > 0.5. The 5300 K stars do not show such a dependence. For example, the $O_2$ value needed for the UV-B flux of 0.82 W/m² marginally increases from an [Fe/H] of −1 to ≈ 0 but decreases for higher [Fe/H] (see black line in Fig. 3o).

Interestingly, the surface UV-B fluxes corresponding to atmospheres with an $O_2$ level higher than 3% monotonically increase with [Fe/H] for all three $T_{eff}$ values considered in this study (Fig. 3j, k, l, with the exception of [Fe/H] ≥ 0.8 and $T_{eff}$ = 5300 K, see Fig. 3l). For example, for the $T_{eff}$ of the Sun the increase of [Fe/H] from −1 to 0.9 doubles

the surface UV-B flux. While surface UV-B fluxes on planets with oxygenated atmospheres may not pose a fatal threat for life on land, their increase with [Fe/H] could negatively affect the evolution of life, especially for planets with low $O_2$ atmospheres, orbiting 5800−6300 K, high-metallicity stars. Since supernovae continuously enrich the Universe with heavy elements over time, stars that form later increasingly contain heavy elements and their planets provide less favourable UV conditions for vegetation and the advancement of complex land life.

While a high stellar metallicity causes UV-B stress for developed life in oxygenated atmospheres, it might be accompanied with faint UV-C radiation levels, being insufficient for the photochemical formation of essential macromolecular building blocks of life in early anoxic atmospheres. For example, it was recently estimated[10] that the average UV actinic flux between 200 and 280 nm should exceed about $6 \times 10^9$ cm⁻² s⁻¹ Å⁻¹ at the planetary surface to allow abiogenesis. Supplementary Figs. 6 and 7 show the dependence of the surface actinic flux on the effective temperature $T_{eff}$ and metallicity [Fe/H] for the 80%-nitrogen and 20%-carbon dioxide atmosphere calculated using the atmospheric transmission dependence on wavelength adopted from Rimmer et al.[10] (we note that in contrast to the oxygenated atmospheres the opacity in anoxic atmospheres is not affected by the net photochemical effect and, thus, is not expected to depend on the stellar spectrum). The actinic flux decreases for cooler and for metal-rich stars, making them less life-friendly. In particular, the actinic flux drops below the Rimmer et al.[10] estimate for 5300 K metal-rich stars.

## Discussion

A key factor for the development of land life is the stability of the atmospheric UV shielding in response to major disturbances that are likely to happen on geological time scales. We performed sensitivity simulations to study the atmospheric effects of potentially catastrophic events such as sudden increases of stellar activity, supernovae and volcanic eruptions. Our model results show that independent of the stellar metallicity such cataclysms do not pose planetary scale, existential threats to life (see methods section Ozone and UV-B response to perturbations and Supplementary Figs. 8–10). While the largest supervolcanoes on Earth have occasionally caused major species extinctions, there are no known examples of such events that have annihilated life[41].

Our results show that from a few percent of $O_2$ upward and for a variety of stellar properties the surface UV exposure on Earth-like planets in habitable zones is likely to sustain the development of land plants that are essential for the further evolution of complex life. It includes a stable oxidation capacity of the atmosphere, which controls greenhouse gases such as $CH_4$, contributing to favourable temperature conditions, while removing hazardous gases that would otherwise reach toxic levels. Paradoxically, whereas stars with higher metallicity, which have appeared later in the history of the Universe, emit less UV radiation, in oxygenated planetary atmospheres the associated stellar radiative spectrum allows less $O_3$ formation, which enhances UV penetration, making the conditions on planets orbiting these stars less friendly for the biosphere on land (except for 5300 K metal-rich stars and low $O_2$ atmospheres). The relatively low UV emission from the high-metallicity stars can also be a hurdle for the origin of first life on planets with anoxic atmospheres.

We thus find that the surface of planets orbiting metal-rich stars is exposed to more intense UV radiation than the surface of planets orbiting metal-poor stars. Therefore planets in the habitable zones of stars with low metallicity are the best targets to search for complex life on land. For the stellar types considered, metallicity has a larger impact on the surface UV than the stellar temperature. The atmospheric oxidation (cleaning) capacity is found to be stable and life-supporting, almost independent of stellar metallicity at an oxygen volume fraction above 1%.

The new generation of radial velocity (RV) spectrometers will be able to measure stellar reflex motion with a precision of 10 cm s$^{-1}$ [42] which suffices to discover Earth-like planets in the habitable zones of Sun-like stars. Detecting Earth-like planets orbiting Sun-like stars is also the main objective of the upcoming PLAnetary Transits and Oscillations (PLATO)[43] of stars space telescope. Our results indicate that to maximise the likelihood of finding signatures of life, planets hosted by low-metallicity stars discovered by these instruments should be priority targets of the follow-up observations with future telescopes[44].

The recently commissioned James Webb Space Telescope (JWST)[45] targets atmospheres of rocky planets around red dwarfs, i.e. stars significantly cooler and smaller than the Sun (since the signal from planets orbiting Sun-like stars is too low to be detected). While planets orbiting red dwarfs are not within the parameter range considered here, one future application of our model will be to simulate the spectral fingerprints of planetary atmospheres[5,6,46–49] observable by JWST as well as anticipated ground-based facilities (like a 2040 s Large Infrared/Optical/Ultraviolet Space Telescope[50]).

## Methods

### Atmospheric model

We applied an updated, global one-dimensional radiative-convective model of a primarily nitrogen/oxygen atmosphere with interactive chemistry[19] for simulations of a wide range of $O_2$ levels and UV-spectra dependent on stellar properties and chemical perturbations. Photolysis rates are calculated with fine spectral resolution (176 wavelength intervals, delta-two-stream method) using equinox conditions and 6 zenith angles for the calculation of daytime average radiation fluxes, taking into account scattering and absorption interactively. Short-lived chemical species are assumed to be in local steady state, while longer-lived ones and chemical families are vertically redistributed by an eddy transport parameterisation. $O_3$ is part of the odd oxygen-family which also contains atomic oxygen in the ground and excited states and radicals of the rate limiting reactions in catalytic destruction cycles of the form

$$O_3 + h\nu \rightarrow O_2 + O \tag{1}$$

$$X + O_3 \rightarrow XO + O_2 \tag{2}$$

$$XO + O \rightarrow X + O_2 \tag{3}$$

X can be NO, OH or halogen atoms. NO and OH are produced in the middle atmosphere from reaction of an excited O-atom from $O_3$ photolysis in the UV-B with $N_2O$ and $H_2O$, respectively. Destruction of odd oxygen also occurs via the Chapman-reaction, which slows down with lower temperatures, e.g. due to radiative cooling by $CO_2$:

$$O + O_3 \rightarrow 2O_2 \tag{4}$$

$O_3$ production is governed by the photolysis of molecular oxygen by UV-radiation with wavelengths shorter than 242 nm, followed by

$$O + O_2 + M \rightarrow O_3 + M \tag{5}$$

with M an arbitrary atmospheric molecule (e.g. $N_2$, $O_2$).

The greenhouse gases $H_2O$, $CO_2$, $CH_4$, $O_3$, and $N_2O$, predominant on Earth, are included in the near and terrestrial infrared radiative transfer calculations. Short-wave radiation reflected to space is determined from the high spectral resolution module used for the photolysis calculations. In the radiation calculations a climatological cloud cover is included and the surface albedo is fixed at conditions on Earth. The surface temperature (Supplementary Fig. 5) is calculated from the radiation budget at the top of the atmosphere, considering

the dependence of the near-infrared part of the stellar spectrum on the effective temperature of the star. It is assumed that the total incoming radiative energy flux at the top of the planetary atmosphere is the same as the solar constant for present day Earth. Lower atmospheric temperatures are calculated from an approximately moist adiabatic lapse rate while middle atmospheric temperatures result from radiative equilibrium. The water vapor feedback for greenhouse warming, i.e. through evaporation from the surface, is included by assuming a fixed relative humidity profile in the lower atmosphere.

The surface boundary conditions for $O_2$ and $CO_2$ on paleo-Earth are established following available geological reconstructions[24,25,27], see Supplementary Table 1. For the atmospheres of exoplanets $CO_2$ volume mixing ratios were fixed to preindustrial (Holocene) conditions on Earth.

Because of the paucity of geological data we used pre-industrial lower boundary conditions for $CH_4$, $N_2O$, $NO_x$, and CO fluxes (230 Tg/yr, 12 Tg/yr, 30 Tg/yr $NO_2$, 1500 Tg/yr, respectively, CO fluxes include oxidation products of volatile organic compounds from land plants) in all simulations. Typical results for the surface are shown in Supplementary Fig. 5. The oxidation of organic substances such as $CH_4$ proceeds via

$$RH + OH \rightarrow R + H_2O \tag{6}$$

where R = $CH_3$ in the case of methane. In the presence of $NO_x$ the further oxidation of R leads to a recycling of OH. The $NO_x$ originates from soil emissions (bacteria), lightning and $N_2O$ breakdown in the middle atmosphere. Primary production of OH proceeds by reaction of an excited oxygen atom from the photolysis of $O_3$ with $H_2O$, and in the case of very low oxygen content by the UV-C photolysis of $H_2O$ (Supplementary Fig. 5). The altitude of the maximum $O_3$ concentration, which decreases with stellar metallicity, shifts downward following the oxygen mixing ratio (Supplementary Fig. 3).

### Ozone and UV-B response to perturbations

In Supplementary Fig. 8b–e we show the $O_3$ column density and surface UV-B levels calculated for the reference simulation, (black), conditions corresponding to a sudden increase of solar magnetic activity (yellow), a supernova explosion (magenta), and a major volcanic eruption (green), to test the stability of atmospheric conditions and compare with previous work. The reference simulation represents the pre-industrial Holocene state of the Earth's atmosphere (i.e. surface fluxes of $CH_4$, $N_2O$, $NO_x$ and CO are set to pre-industrial levels) under a stellar radiation intensity according to the minimum of solar cycle 22.

The high activity simulations are forced by the solar spectrum calculated for the Sun with an S-index value of 0.25, which corresponds to a magnetic activity level five times higher than during the maximum of solar cycle 22. This is motivated by the recent analysis of data from the Kepler space telescope[51] and Gaia space observatory[52] which hinted that the Sun might go through occasional epochs of high magnetic activity[22,53]. The intensification of stellar activity results in an increase of UV emission (Supplementary Fig. 11) that is stronger for radiation with shorter wavelength[23]. Consequently, the $O_3$ production rate increases more strongly than the destruction rate resulting in an increase of the total $O_3$ column density (Supplementary Fig. 8b). This result agrees with previous studies of the $O_3$ response to the solar activity cycle[54,2] which also reported a positive correlation between $O_3$ column density and solar magnetic activity. The increase in $O_3$ enhances the protection from UV-B which overcompensates the increase of the solar UV-B for an $O_2$ content larger than 3% and damps it for smaller $O_2$ amounts. Consequently, the increase of stellar magnetic activity does not pose a serious threat to the biosphere in terms of UV-B exposure.

Another potential perturbation is a supernova explosion of a neighbouring star[14,55] that increases the flux of charged particles

entering the planetary atmosphere[20]. On Earth, most particles are deflected by the heliospheric and geo-magnetic fields, but a fraction enters the atmosphere and produces $NO_x$ from molecular $N_2$ and $O_2$. While $NO_x$ is capable of destroying $O_3$ in catalytic cycles in the upper and middle atmosphere, it can lead to $O_3$ production in the lower atmosphere[14,21,55]. To simulate the response of the planetary atmosphere to supernovae we used a previous approach to estimate terrestrial $O_3$ depletion[20]. We increased the NO production by charged particles by a factor of 100 relative to the reference simulation (which roughly corresponds to the effect of supernova exposure at 10 pc). The resulting $O_3$ column density and surface UV-B are shown in Supplementary Fig. 8 by the magenta curves. We find that the supernovae impact on $O_3$ column density and surface UV-B is weak, in agreement with previous work (independent of metallicity and $O_2$ level). In oxygenated atmospheres ($O_2$ level more than ~2%) supernovae lead to $O_3$ destruction and an increase of surface UV-B. When the $O_2$ content is relatively low, the additional production of $O_3$ at lower altitudes in the atmosphere helps decrease the UV-B penetration (Supplementary Figs. 9 and 10 for $T_{eff}$ of 5300 K and 6300 K, respectively).

Major volcanic eruptions can potentially damage the ozone layer[13,56]. The release of sulfur dioxide leads to the formation of sulfate particles which facilitate heterogeneous reactions that activate halogen-containing radicals that cause $O_3$ destruction[57,58]. Furthermore, volcanic eruptions can directly inject halogens into the ozone layer[58,59]. To estimate potential volcanic impacts we increased the amount of sulfate particles by an order of magnitude. Reactive chlorine and bromine components were increased by injecting 3Tg HCl and 30Gg HBr per year into the lower stratosphere. This scenario mimics the end-Permian eruption of the Siberian Traps[41]. The resulting $O_3$ column density and surface UV-B changes are shown in Supplementary Fig. 8 by the green curves. The $O_3$-depleting effect of volcanism can be strong but weakens when the $O_2$ content reaches about 10%. The volcanic impact is slightly weaker for metal-poor stars but does not drop with $O_2$ content (see Supplementary Figs. 9 and 10 for $T_{eff}$ of 5300 and 6300 K, respectively).

## Stellar spectra

The stellar spectra for different $T_{eff}$ and metallicity values [Fe/H] have been calculated with the MPS-ATLAS code[17]. The metallicity is defined relative to the solar metallicity:

$$[\mathrm{Fe/H}] = \log\left(N_{Fe}/N_{H}\right)_{star} - \log\left(N_{Fe}/N_{H}\right)_{\odot} \qquad (7)$$

where $N_{Fe}$ and $N_H$ are the numbers of Fe and H atoms per volume unit. The chemical composition was taken from Asplund et al.[60].

For each set of $T_{eff}$ and [Fe/H] values we first computed the stellar model atmosphere using the dependence[61] of mixing-length parameters on $T_{eff}$ and [Fe/H] normalised to return a solar value of 1.25[62] for solar $T_{eff}$ and [Fe/H]. We have accounted for the line blanketing using the Opacity Distribution Functions which were computed utilising more than 100 millions of atomic and molecular lines whose spectral shape was calculated for a turbulent velocity value of 2 km s$^{-1}$ [17,62]. We have also accounted for the missing UV opacity following the approach by Shapiro et al.[63].

The emergent stellar spectrum was calculated on 176 wavelength intervals exactly corresponding to our radiation code in the planetary atmospheric model (see section Atmospheric model) for 24 disk positions and then integrated over the full stellar disk. We have considered the same sources of opacity as for computing the stellar atmospheric models. In particular, a large number of spectral lines is needed for realistic calculations of the line haze which dominates the UV opacity.

Finally, the spectra have been scaled to preserve the amount of total radiative energy the planet receives. The spectra of the active Sun have been calculated following the approach by Shapiro et al.[23] as a function of the S-index which is a standard measure of stellar activity and is proportional to the ratio of the fluxes in Ca II H and K lines and the nearby continuum[64].

## Data availability

The data that support the findings of this study are available from the corresponding author upon request. Stellar normalized photon fluxes in the 176 spectral intervals for all metallicities and the three effective temperatures are available in source data file input_flux_photons_{T}.txt. Source data are provided with this paper. The data sets generated during the current study are available in the Edmond repository, https://doi.org/10.17617/3.WGVDYV. Source data are provided with this paper.

## Code availability

The atmospheric and MPS-ATLAS codes used in the current study are available from the corresponding author on reasonable request.

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

## Acknowledgements

A.V.S. and L.G. acknowledge funding from the Max Planck Society (grant "Preparations for PLATO science"). N.K. and L.G. acknowledge funding from the German Aerospace Center (DLR FKZ 50OP1902 "PLATO Data Center"). A.I.S. and V.W. were funded by the European Research Council

(ERC) under the European Union's Horizon 2020 research and innovation program (Grant no. 715947).

## Author contributions

A.V.S. conducted the simulations and coordinated the writing of the manuscript. C.B. provided and adapted the model used for calculating the planetary atmosphere. C.B., K.K., B.S., and J.L. provided expertise on atmospheric physics and chemistry. N.K., V.W., and A.I.S. calculated the stellar spectra. A.I.S., L.G., and S.K.S. provided the expertise on stellar physics. A.V.S., A.I.S., C.B., K.K., and B.S. prepared the first draft. All authors contributed to the final manuscript. All authors discussed the results and reviewed the manuscript.

## Funding

## Competing interests

The authors declare no competing interests.
