## [Peer Review File · Nature Communications]

REVIEWERS' COMMENTS

Reviewer #1 (Remarks to the Author):

The paper by Shapiro et al. presents new results that relate the stellar metallicity and the surface UV environment considering planets that already have life producing O₂. The relationship of stellar UV, O₂, O₃ and the UV that reaches the planetary surface is not new but the relation of all these parameters with the stellar metallicity has not been explored.

The authors have satisfactorily addressed all my comments and the paper has been improved accordingly. I just the following minor comments:

1. Kozakis et al. (AA, 2022) have reach similar conclusions about the O₂-O₃-UV relationships, I suggest including this paper in the references. Particularly, the statement in lines 125-127 is what Kozakis et al. have found too.
2. Please provide a reference for the statement in lines 72-73.
3. The wording "column O₃" is not used in the planetary atmosphere's literature. It would be more appropriate "O₃ column density"

Reviewer #2 (Remarks to the Author):

The authors have addressed my major concerns. I have no further substantive feedback. I congratulate the authors on a careful and creative study.

Reviewers Comments:

Reviewer #1 (Remarks to the Author):

SUMMARY

Reviewer: The paper by Shapiro et al. presents new results that relate the stellar metallicity and the surface UV environment considering planets that already have life producing O₂. The relationship of stellar UV, O₂, O₃ and the UV that reaches the planetary surface in not new but the relation of all these parameters with the stellar metallicity has not been explored. The authors have satisfactorily addressed all my comments and the paper has been improved accordingly. I just the following minor comments:

Our answer: We thank the reviewer for the positive evaluation of our manuscript and for the useful suggestions. We have revised the manuscript following the recommendations and give a point-by-point reply below.

Reviewer: 1. Kozakis et al. (AA, 2022) have reach similar conclusions about the O₂-O₃-UV relationships, I suggest including this paper in the references. Particularly, the statement in lines 125-127 is what Kozakis et al. have found too.

Our answer: We thank the reviewer for the suggestion. We included the reference to Kozakis et al. (AA, 2022).

Reviewer: 2. Please provide a reference for the statement in lines 72-73.

Our answer: We added the reference to Brasseur&Solomon (2005) that presents an extensive description of radiative processes on the Earth's atmosphere.

Reviewer: 3. The wording "column O₃" is not used in the planetary atmosphere's literature. It would be more appropriate "O₃ column density"

Our answer: We thank the reviewer for this helpful comment. We corrected the manuscript following this suggestion.

Reviewer #2 (Comments for the Author):

Reviewer: The authors have addressed my major concerns. I have no further substantive feedback. I congratulate the authors on a careful and creative study.

Our answer: We thank the reviewer for this kind words and for improving this manuscript.